# Relationship between Shift Type and Voluntary Exercise Training in South Korean Firefighters

**DOI:** 10.3390/ijerph17030728

**Published:** 2020-01-22

**Authors:** Seiyeong Park, Junhye Kwon, Kyoungmin Noh, Chung Gun Lee, Wook Song, Jung-jun Park, Han-joon Lee, Dong-il Seo, Hyun Joo Kang, Yeon Soon Ahn

**Affiliations:** 1Department of Physical Education, College of Education, Seoul National University, Seoul 08826, Korea; pseiy09@snu.ac.kr (S.P.); 2017_25059@snu.ac.kr (J.K.); nkm0909@snu.ac.kr (K.N.); songw3@snu.ac.kr (W.S.); 2Institute of Sport Science, Seoul National University, Seoul 08826, Korea; 3Institute on Aging, Seoul National University, Seoul 08826, Korea; 4School of Sport Science, Pusan National University, Pusan 46241, Korea; jjparkpnu@pusan.ac.kr; 5School of Sport Science, Ulsan University, Ulsan 44610, Korea; hanjoon@ulsan.ac.kr; 6Department of Sport Science, College of Liberal Arts, Dongguk University, Gyeongju 38066, Korea; seodi74@dongguk.ac.kr; 7Department of Sport Medicine, College of Natural Science, Soonchunhyang University, Asan 31538, Korea; violethjk@naver.com; 8Department of Preventive Medicine and Genomic Cohort Institute, Yonsei Wonju College of Medicine, Yonsei University, Wonju 26426, Korea; ysahn1203@gmail.com

**Keywords:** firefighter, exercise training, shift work

## Abstract

Background: According to the National Fire Agency, 69% of South Korean firefighters prefer the three circuit system. Since the three circuit system involves working for 24 h consecutively, it could reduce work performance of firefighters and their participation in exercise training (ET) and fitness levels could be affected by shift type. The present study examined the impact of shift type on ET and identified the interaction effect between shift type and city size on ET among South Korean firefighters. Methods: A series of logistic regression models were used to analyze the data collected from incumbent outside workers among Korean firefighters (N = 5196) in 2017. Results: Firefighters under the 3 circuit system participated in ET less frequently than did those under the 21 circuit system, and the difference was greater in large cities than in small towns. This could be because of the greater opportunity to participate in leisure activities in large cities, or because participating in ET is easier for firefighters in small towns, who tend to be less busy than those in large cities. Conclusions: The three circuit system is not feasible in the present situation in South Korea. Therefore, an environment suitable for the three circuit system should be established.

## 1. Introduction

Firefighters are responsible for protecting the safety of the public. They may be on-call for 24 h straight and are required to perform extreme physical activities. It is thus important for firefighters to maintain a high level of fitness so that they can carry victims or use heavy equipment [1,2,3]. However, the prevalence of job and lifestyle-related chronic diseases among South Korean firefighters is four to seven-fold higher than that in the general population, and their fitness level is lower than that of U.S. firefighters [4,5,6,7]. It reflects the fact that South Korean firefighters are either unaware of the importance of exercise training (ET) or have insufficient opportunities for ET [8,9].

The National Fire Protection Association (NFPA) promulgated ‘NFPA 1583 (standard for health-related fitness programs for fire department members)’ to improve the performance of U.S. firefighters [10]. It requires that firefighters voluntarily participate in ET, and that ET programs be provided by the independent Joint Labor Management Wellness–Fitness Initiative (WFI) under the International Association of Fire Fighters. Moreover, the WFI educates expert fire service peer fitness trainers (https://www.acefitness.org/fitness-certifications), performs fitness evaluations, and designs ET [6]. Indeed, firefighters can return to work only after undergoing a fitness test [11].

Firefighters in the UK are required to participate in an exercise program and to take an examination at 6 month intervals. If their fitness level is below the acceptable standard, UK firefighters are removed from duty and are required to complete a supplemental course to improve their fitness level; moreover, if a serious accident occurs, UK firefighters can be sued and accused of manslaughter because of their low fitness level, which hampers their ability to protect life and property [12]. Organizational efforts for managing the fitness level of firefighters are related to the occupational ability of firefighters and public safety [10,13,14,15]. However, in South Korea, ET is not mandatory. A few provisions have attempted to enforce ET by Korean firefighters, but the implementation of these provisions is at the whim of the heads of the firefighting agencies [16,17]. Although South Korean firefighters are allotted time for ET daily, participation is voluntary.

By 2017, 97% of South Korean firefighters were under the three shift system (three groups, two circuits). In South Korea, the three shift system was trialed in 2009 as an improvement over the two shift system. The three shift system involves 6 circuits (day–day–night–night–off duty–off duty), 9 circuits (day–day–day–night–off duty–night–off duty–night–off duty), 21 circuits (day–day–day–day–day–off duty–off duty–night–off duty–night–off duty–night–off duty–duty–off duty–night–off duty–night–off duty–duty–off duty), or 3 circuits (duty–off duty–off duty). This change has reduced the number of work hours per week from 84 to 56; however, it was developed for advanced nations and is not suitable for South Korea. South Korean firefighting agencies are facing personnel shortages. The number of people per firefighter is 1181 in South Korea, which is higher than that in other countries that use the three shift system, such as Hong Kong (787 per person), Japan (799 per person), and the United States (936 per person). This increases the rate of suicide among firefighters and reduces their performance compared with the two shift system [18,19].

According to the National Fire Agency, 69% of South Korean firefighters prefer the three circuit system [20]. The three circuit system is used by firefighters in the United States, Singapore, France, and Germany. Unlike the three shift system in South Korea, this system in other nations involves 48 h of rest after 24 h of work. The three circuit system allows fitness recovery of the firefighters, reduces the commute time, and maintains a regular life rhythm because an overnight shift is followed by 2 days of rest [21,22]. Therefore, it is being piloted in 1486 (4.2%) firefighters in seven regions of South Korea as of April 2017 [23]. However, because the three circuit system involves working for 24 h consecutively, it could reduce work performance, increase fatigue, and cause sensitivity, insomnia, and obesity [24,25].

Before changing shift system in South Korea, we need to contemplate because participation in ET by firefighters differs according to shift type [26]. According to the 2017 Employment Impact Assessment, the best time and place for ET among South Korean firefighters is ‘at work within working hours’, and ‘individual leisure time’ was the least time due to their job [18]. Firefighters prefer to spend their time on leisure activities likely because ET is a work-related activity. Firefighters in bureaucratic organizations may be influenced by their colleagues or the work atmosphere [27,28]. In South Korea, participation in ET by firefighters is voluntary; the only motivation to participate is the annual physical fitness test [29]. Thus, transition to the three circuit system in South Korea would reduce participation in ET and thereby increase the risk of physical and mental deterioration and accidents while on duty. In other words, adoption of the three circuit system by South Korean firefighters would threaten the safety of not only themselves but also their colleagues and citizens.

There has been no study of the effect of the three circuit system on voluntary participation in ET among South Korean firefighters before its implementation. Therefore, we investigated the effect of two different shift types (i.e., 3 circuit and 21 circuit) on voluntary participation in ET by Korean firefighters. We also investigated the effect of city size on the relationship between shift type and voluntary ET participation because the three circuit system is being piloted among South Korean firefighters only in small towns, in which the intensity of work is lower than that in large cities.

## 2. Materials and Methods

### 2.1. Data

We used data from an online survey conducted in 2017 by the Ministry of Public Safety and Security (currently the National Fire Agency) in South Korea. The survey was performed to improve the safety and health of South Korean firefighters based on their work-related characteristics. The subjects of the survey were about 40,000 South Korean firefighters nationwide. They were informed in writing of the purpose of the study, the research plan, privacy, and data confidentiality. Those who did not wish to participate in the survey were excluded from the study. The questionnaire comprised questions on general and professional characteristics, such as shift work, shift–work circuit, and job type, and ET participation, environment, and preferences. On average, completing the questionnaire took about 45 min.

### 2.2. Measures

For firefighters, physical fitness is defined as the ability to successfully perform a task in the event of a fire such as saving a victim, pulling a hose, climbing a stairway, supporting weight, or preparing for an attack [27]. ET is defined as physical activity that promotes the ability of the firefighter to perform their duties in the event of an incident. It was assessed by the following questions: “Do you participate in ET to improve fitness?”, “How many times a week do you participate in ET?” (e.g., once a week), “How many minutes do you perform ET at once?” (e.g., 60 min at once), and “What intensity do you perform ET?” Among the responses for shift type, 3 circuit which people are interested in and 21 circuit which is adopted in Seoul were chosen to analyze shift type. In South Korea, the three circuit system is being trialed, and 90% of the firefighters involved live in small towns. For this reason, city size was included in the model to control for its influence on participation in ET, based on the 2018 Guidelines for Establishment and Operation of Life-Friendly Centers [30].

### 2.3. Statistical Analysis

We performed logistic regression analysis of the data from 5196 South Korean firefighters to evaluate the influence of shift type on ET using SAS version 9.4 (SAS Institute Inc., SAS Campus Drive, Cary, NC, USA). We only included data from firefighters working under 3- and 21 circuit system for the analysis. All of them were outside workers (i.e., firefighting, first aid, and rescue) because we outside workers are required to have higher levels of fitness to succeed physically demanding tasks compared to inside workers. Data were divided according to the degree of participation in moderate-to-vigorous ET for >150 min/week based on the physical activity guidelines for American adults [31]. The regression model included gender, age, job, education level, smoking and drinking, shift type, and city size. Individual demographic variables such as gender, age, job, and education level were controlled for in model 1. Demographic and health behavioral variables such as smoking and drinking were controlled for in model 2. Demographic, health behavioral, and job-related variables (such as job and shift type) were controlled for in model 3, and city size was added in model 4. Model 5 was the full model and included the interaction term between shift types and city size.

The reference for ET was firefighters who participate in ET more than 150 min per week, the reference for gender was male, the reference for shift type was the three circuit system, and the reference for city size was large cities. Age was centered on the mean age of all participants for ease of interpretation.

## 3. Results

### 3.1. Descriptive Statistics

As shown in Table 1, the 5196 firefighters comprised 4803 (92.44%) males and 393 (7.56%) females. The mean age was 38.14 (±8.68) years. Among the firefighters, 517 (9.95%) and 4679 (90.05%) worked under the 3- and 21 circuit systems, respectively, and 2324 (46%) participated in ET for >150 min per week.

### 3.2. Participation in Exercise Training

Table 2 shows the logistic regression analysis results for the effects of the variables on participation in ET among South Korean firefighters. In model 1, gender (Coef = 0.811, SE = 0.123, *p* < 0.001), age (Coef = −0.012, SE = 0.005, *p* < 0.05), and education level (college: Coef = −0.342, SE = 0.084, *p* < 0.001; university: Coef = −0.342, SE = 0.079, *p* < 0.001) significantly impacted participation in ET. In model 2, smoking significantly affected participation in ET (Coef = −0.345, SE = 0.066, *p* < 0.001). Shift type did not impact ET participation significantly in model 3; however, it had a significant effect after adjusting for city size, in model 4 (Coef = −0.326, SE = 0.103, *p* < 0.05). Also, in model 5, the interaction term between city size and shift type exerted a significant effect on ET participation (Coef = −1.146, SE = 0.394, *p* < 0.01).

Figure 1 shows the interaction effect between shift types and city size on participation in ET. Under 3 circuit system, the effect of city size on the participation in ET became significantly larger than under 21 circuit system. In other words, the significant change in voluntary ET participation according to shift type is greater in large cities but not in small towns.

## 4. Discussion

We investigated the effect of shift type on voluntary ET among South Korean firefighters. It is expected that the three circuit system will increase ET participation; however, this has not been demonstrated [20]. We compared the effect of the 3 and 21 circuit systems on participation in ET and included the interaction between shift type and city size. The results showed that ET was engaged in more frequently by males than females, and by college or university graduates than high school graduates. According to our results, sex has the strongest effect on ET among South Korean firefighters. In South Korea, male firefighters might be more active in ET participation than female firefighters because the current criteria on fitness levels of firefighters is unequal by gender. However, female firefighters are required to perform physically demanding tasks without discrimination so that they need to be evaluated by higher criteria and to improve their fitness levels [4]. Smokers participate in ET more than non-smokers in our results and previous studies [32,33,34,35,36,37]. These results are contrary to general expectations but sometimes people who participate in less physical activity overestimate and people who are more engaged in physical activity underestimate physical activity [38,39,40,41]. We found that younger firefighters tend to participate in ET more than older firefighters. In contrast, in the general public, those aged 50–60 years participate in ET more frequently than those aged 30–40 years in South Korea [42]. This is probably because more junior firefighters tend to be younger, and their duties require a high level of fitness. Moreover, firefighters under the 21 circuit system participate in ET more frequently than those under the 3 circuit system. Firefighters may consider participation in ET is a work-related activity, and thus firefighters choose to spend their off duty days resting or being with family or friends more often than participating in ET [18]. Thus, shift type influences voluntary participation in ET by South Korean firefighters.

Regarding the interaction effect between shift type and city size on participation in voluntary ET, we can find the expanded differences in the effects of city size on ET under the 3 circuit system. The reason why the effect of the types of city on participation in ET is greater under 3 circuit is that the firefighters under 3 circuit system mainly live in small towns. Firefighters living in small towns have less opportunities for leisure activity compared with firefighters living in large cities. Furthermore, because there are fewer dispatches in small towns than in large cities, firefighters in small towns would be able to participate in ET during working hours.

This is the first study that examines a relationship between shift type and participation in voluntary ET among South Korean firefighters. We tried to identify whether changing shift type into 3 circuit from 21 circuit make firefighters more participate in ET. Participating in ET of firefighters is related to health and safety of the publics as well as firefighters’ physical ability. However, this study is not without several limitations. First, the results of the analysis in this study could not be applied to the general population. For firefighters, our target population, participation in ET would be considered as a job-related activity to enhance their job performance. Therefore, they would have different aims to participate in ET compared with general population. Second, the city size variable could not exactly represent the job environment of firefighters like the numbers of dispatches. We suggest identifying physical, social, and cultural environment surrounding South Korean firefighters for future studies. Third, the numbers of participants between two shift types were inconsistent in our data because the shift types are differently adopted in each South Korean region. In recent years, 21 circuit has been used in large cities, comparing that 3 circuit has been operated in small towns in South Korea. Therefore, we controlled the effect of city size by using an interaction term in Model 5. Forth, we could not decide the causal relationship between variables because the present study is a cross-sectional study. Thus, readers of this paper are required to interpret with some cautions. Finally, all the constructs were self-reported in this study. It could make problems in the way that it can bring response bias or recall bias. Although the present study has some limitations, the results of our study may contribute to the literature by giving valuable information related to ET participation of South Korean firefighters.

## 5. Conclusions

It is important for firefighters to participate in ET because a high level of fitness among firefighters is directly connected to the public health and safety. The present study has identified Korean firefighters under the 3 circuit system participate in ET less than those under the 21 circuit system. In the current fire situation, if firefighters have more leisure time by changing their shift system to 3 circuit, they would spend time enjoying their leisure activities or family time because ET might be considered as a work-related activity for them. Therefore, a change to the 3 circuit system will reduce voluntary participation in ET among South Korean firefighters and then it would endanger public safety. More interestingly, the effect of shift type on ET participation is represented differently according to city size among South Korean firefighters. Understanding situations in other countries that accept the 3 circuit system for firefighters would help to find a suitable system for South Korean firefighters and to identify circumstances for implementation of the 3 circuit system in South Korea.

## Figures and Tables

**Figure 1 ijerph-17-00728-f001:**
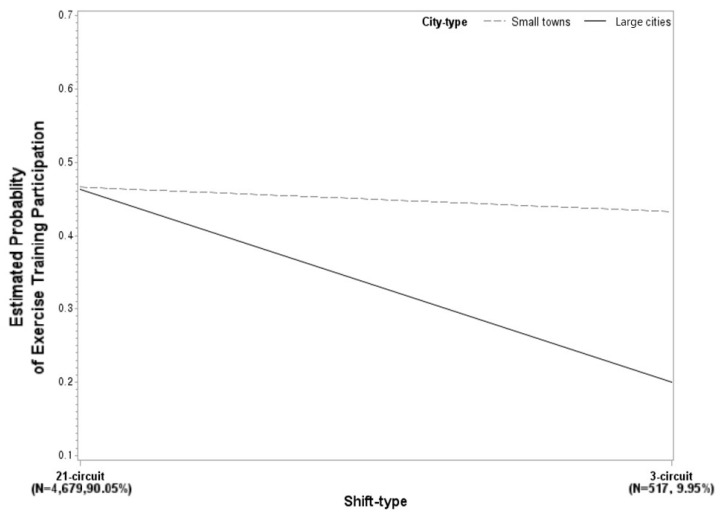
The interaction effect between shift type and city size on exercise training participation.

**Table 1 ijerph-17-00728-t001:** Characteristics of firefighters.

Characteristics	Category	N	%
Number of Participants	Total	5196	
Sex	Male	4803	92.44
	Female	393	7.56
Age (Mean ± SD)		38.14 ± 8.68	
Job position ^1^	Low	2594	56.85
	Middle	2207	42.48
	High	35	0.67
Education	High school	997	19.20
	College	1733	33.37
	University	2277	43.85
	Graduate school	72	1.39
Drinking	Yes	3823	73.62
	No	1370	26.38
Smoking	Yes	1462	28.15
	No	3731	71.85
Shift-type	3 circuit	517	9.95
	21 circuit	4679	90.05
City size	Large city	2038	42.49
	Small town	2758	57.51
Exercise training	≥150	2324	46.37
(min/week)	<150	2688	53.63

^1^ Low job position includes firefighter, senior fireman; Middle job position includes fire sergeant, fire lieutenant, fire captain; High job position includes assistant fire chief, fire chief, above fire sub-deputy chief.

**Table 2 ijerph-17-00728-t002:** Logistic regression models examining the relationships between variables and exercise training (N = 4653).

Characteristics	Model 1		Model 2		Model 3		Model 4		Model 5	
	Coef	SE	Coef	SE	Coef	SE	Coef	SE	Coef	SE
Intercept	−0.698 ***	0.145	−0.746 ***	0.152	−0.727 ***	0.152	−0.697 ***	0.162	−0.716 ***	0.162
Sex										
Men (ref)	-	-	-	-	-	-	-	-	-	-
Women	0.811 ***	0.123	0.904 ***	0.125	0.902 ***	0.125	0.897 ***	0.131	0.900 ***	0.131
Age	−0.012 *	0.005	−0.012 *	0.005	−0.012 *	0.005	−0.012 *	0.006	−0.011 *	0.006
Education										
High school (ref)	-	-	-	-	-	-	-	-	-	-
College	−0.342 ***	0.084	−0.347 ***	0.084	−0.348 ***	0.084	−0.350 ***	0.087	−0.354 ***	0.087
University	−0.342 ***	0.079	−0.351 ***	0.080	−0.353 ***	0.080	−0.343 ***	0.082	−0.346 ***	0.082
Graduate school	−0.375	0.259	−0.398	0.260	−0.414	0.261	−0.382	0.277	−0.390	0.277
Job position										
Low (ref)	-	-	-	-	-	-	-	-	-	-
Middle	0.146	0.093	0.128	0.094	0.127	0.094	0.105	0.098	0.109	0.098
High	0.293	0.351	0.265	0.352	0.263	0.352	0.140	0.364	0.134	0.364
Drinking										
Yes (ref)			-	-	-	-	-	-	-	-
No			0.101	0.067	0.101	0.067	0.080	0.070	0.081	0.070
Smoking										
Yes (ref)			-	-	-	-	-	-	-	-
No			−0.345 ***	0.066	−0.344 ***	0.066	−0.325 ***	0.070	−0.325 ***	0.070
Shift-type										
3 circuit (ref)					-	-	-	-	-	-
21 circuit					−0.164	0.097	−0.326 *	0.103	−0.108	0.109
City size										
Large city (ref)							-	-	-	-
Small town							−0.046	0.063	−0.009	0.064
Interaction										
Shift-type × City size									−1.146 **	0.394

Note: Model 1 is the sex, age, education, and job-position controlled model. Model 2 is the drinking, smoking and sex/age/education/job-position controlled model; Model 3 is the shift-type, drinking/smoking and sex/age/education/job-position controlled model; Model 4 is the city size, shift-type, drinking/smoking and sex/age/education/job-position controlled model; and Model 5 is the full model where interaction term was entered into Model 4. Coef = logistic coefficient; ref = reference category. * *p* < 0.05, ** *p* < 0.01, *** *p* < 0.001.

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
