# Peer review of "Relationship between Shift Type and Voluntary Exercise Training in South Korean Firefighters"

_ijerph, 2020, doi:10.3390/ijerph17030728_

Round 1

Reviewer 1 Report

This is a good research study examining the effect of shift work patterns and city size on exercise training.

There are a few suggestions, but on the whole this was well written and presented. 

Line 21 – Because – please use another word at the start of the sentence.

If they work for 24 hours consecutively how much rest time do they get following this?

Line 25 regression models are pleural so please use the word were instead of was

Line 183 do you mean 50-60 years and 30-40 years and not 5-60 years and 3-40 years?

Author Response

Line 21 – Because – please use another word at the start of the sentence.

Thank you for your comment. We modified it based on your comment. The change is yellow-highlighted.

If they work for 24 hours consecutively how much rest time do they get following this?

They can take a rest for 48 hours after working 24hours under 3 circuit system in South Korea.

Line 25 regression models are pleural so please use the word were instead of was

Thank you for your comment. We amended it based on your comment. The change is yellow-highlighted.

Line 183 do you mean 50-60 years and 30-40 years and not 5-60 years and 3-40 years?

Thank you for your comment. We corrected it based on your comment. The change is yellow-highlighted.

Reviewer 2 Report

I have been reading your paper "Relationship between Shift Type and Voluntary Exercise Training in South Korea Firefighters" explaining the importance of Voluntary Exercise Training and shift Type. In general, the paper is important for the firefighting activity, but it has to be better described and a higher focus of the firefighters are necessary.

The introduction is well described but the method, results and discussion have to be a bit more explained in order to help the reader to understand the results and not to do the wrong conclusion.

Specific comments:

Page 3, line 94: from what I understand in the result section you do study differences between the 3 circuit system and the 21 circuit system (in the introduction you do explain a lot of different system) and that should be pointed out int the aim.

Page 3, line 104: The response rate is rather low (only about 12 %). There is no information about inclusion or exclusion. I understand the text as those who did not answer the survey were excluded. Instead they should be non responders and the low response rate have to be discussed. The low response rate is important for the conclusion. OR did you only include 21- and 3 circuit system and others were excluded? If so, you have to explain the inclusion- and exclusion criterias.

Page 3, line 115: Please report possible answers on each question.

Table 1: Data are skewed based on how many included participants on the different shift systems, that should also be discussed.

Page 3, line 123-136: Please include the statistical program used and coding of data.

Page 4, Table 2: It should be proper to include the number om respondents for small towns and also large cities and also somewhere in the method present the difference. That is important for the results and for Figure 2.

Page 4, line 150: " shift type did not impact ET participation significantly in model 3, however it had a significant effect after adjusting for city size in model 4" Have you thought about the fact that the overall model did not improve when adding shift type and city size?

Page 4, Figure 1: The Figure is incomplete- please explain the y-axis and the x-axis (N? %?).

Page 7, line 181: Interesting that you found that smokers participate in ET more than non-smokers. Often persons with low physical activity overestimate the physical activity and persons with high physical activity underestimate the physical activity- pleas find references and include in the discussion.

I can not see any discussion about the fact that sex was the variable affecting the model most?

Page 7, line 183: "in contrast, in the general public, those aged 5-60 years participate in ET more frequently than those aged 3-40 years" First, I can not see any referencene, please include. Second, do you mean 50 to 60 and 30-40?

Page 7, line 202: With 40.000 within the fire department asked to answer to survey (if I have not understood the text) do you think it could be a general result for the fire department? I can see that you mentioned it in the discussion, but I would like to see a broader discussion.

Page 7, line 202: This is a good time to discuss and present the low response of the questionnaire. Also, to discuss the response rate of only firefighters since the text in general is focusing of their profession and their performance.

Page 7, I think the authors instead should focus on firefighters in the paper and in the conclusion in order to give a input to the firefighting job and performance. The heading is focusing of firefighters but the paper and the conclusion is not.

Author Response

Page 3, line 94: from what I understand in the result section you do study differences between the 3 circuit system and the 21 circuit system (in the introduction you do explain a lot of different system) and that should be pointed out int the aim.

Thank you for your comment. We agree with your idea. Therefore, we specified shift types based on your comment. The changes are yellow-highlighted.

Page 3, line 104: The response rate is rather low (only about 12 %). There is no information about inclusion or exclusion. I understand the text as those who did not answer the survey were excluded. Instead they should be non responders and the low response rate have to be discussed. The low response rate is important for the conclusion. OR did you only include 21- and 3 circuit system and others were excluded? If so, you have to explain the inclusion- and exclusion criterias.

Thank you for your comment. We only included data from firefighters working under 3- and 21 circuit system for the analysis. Also, all of them were outside workers (i.e., firefighting, first aid, rescue) because outside workers are required to have higher levels of fitness to succeed in physically demanding tasks compared to inside workers. We explained the inclusion criteria in the Methods. The changes are yellow-highlighted.

Page 3, line 115: Please report possible answers on each question.

Thank you for your valuable comment. We reported possible answers based on your opinion. The changes are yellow-highlighted.

Table 1: Data are skewed based on how many included participants on the different shift systems, that should also be discussed.

The reason why the numbers of participants are different was due to the inconsistency of shift types between regions in Korea. We included the explanation in the Discussion based on your comment. The changes are yellow-highlighted.

Page 3, line 123-136: Please include the statistical program used and coding of data.

We included a phrase to report the statistical program we used. The changes are yellow-highlighted.

Page 4, Table 2: It should be proper to include the number om respondents for small towns and also large cities and also somewhere in the method present the difference. That is important for the results and for Figure 2.

Thank you for your comment. We included the number of respondents for small towns and large cities in Table 2 based on your comment.

Page 4, line 150: " shift type did not impact ET participation significantly in model 3, however it had a significant effect after adjusting for city size in model 4" Have you thought about the fact that the overall model did not improve when adding shift type and city size?

Thank you for your valuable comment. However, assessing model fits was not important for our results because we wanted to control city size just to identify the effect of city size on the relationship between shift type and ET.

Page 4, Figure 1: The Figure is incomplete- please explain the y-axis and the x-axis (N? %?).

Thank you for your comment. Two shift types used in the analysis were shown on the x-axis and the estimated probability of ET participation followed the y-axis. We revised Figure 1 based on your comment.

Page 7, line 181: Interesting that you found that smokers participate in ET more than non-smokers. Often persons with low physical activity overestimate the physical activity and persons with high physical activity underestimate the physical activity- pleas find references and include in the discussion.

Thank you for your comment. We found references and included that based on your comment. The changes are yellow-highlighted.

I can not see any discussion about the fact that sex was the variable affecting the model most?

Thank you for your comment. We thought the reason why sex most strongly affects to the model was that Korean firefighters have different standards for fitness levels between genders. We included additional explanations related to that fact in the Discussion based on your comment.

Page 7, line 183: "in contrast, in the general public, those aged 5-60 years participate in ET more frequently than those aged 3-40 years" First, I can not see any referencene, please include. Second, do you mean 50 to 60 and 30-40?

Thank you for your comment. We included a reference reporting this fact and modified them properly based on your opinion.

Page 7, line 202: With 40.000 within the fire department asked to answer to survey (if I have not understood the text) do you think it could be a general result for the fire department? I can see that you mentioned it in the discussion, but I would like to see a broader discussion.

Our results could be generalized for the fire department because data used in our analysis were collected by an authoritative national census on Korean firefighters. However, we analyzed data only from outside workers (i.e., firefighting, first-aid, and rescue) because we thought they are fitter for the aims of our study. We suggest that future researchers collect and analyze data of both outside workers and inside workers (e.g., driving, fire investigation, administration) so that they obtain more exact knowledge about the situations surrounding shift systems and exercise behavior of firefighters.

Page 7, line 202: This is a good time to discuss and present the low response of the questionnaire. Also, to discuss the response rate of only firefighters since the text in general is focusing of their profession and their performance.

Thank you for your comment. As we answered former questions, we used data only from outside workers for analysis.

Page 7, I think the authors instead should focus on firefighters in the paper and in the conclusion in order to give a input to the firefighting job and performance. The heading is focusing of firefighters but the paper and the conclusion is not.

Thank you for your comment. We added sentences in Conclusions based on your comment. The changes are yellow-highlighted.

Round 2

Reviewer 2 Report

Hi!

Thank you for the revised version of the manuscript. The paper has improved but there are still some work to do.

In the aim you are writing “Therefore, we investigated the effect of two different shift types (i.e., 3 circuit and 21 circuit) on voluntary participation in ET by Korean firefighters” And in the method “We only included data from firefighters working under 3- and 21 circuit system for the analysis. Also, all of them were outside workers (i.e., firefighting, first aid, rescue) because we outside workers are required to have higher 128 levels of fitness to succeed physically demanding tasks compared to inside workers” Looking at Table 1 you can see the education level and you have explained in text that “High job position includes assistant fire chief, fire chief, above fire sub-deputy chief” Are the high job positions really outside workers?

Page 3, line 115: If Yes and No are the only possible answers e.g should be excluded.

Page 3, line 117: Regarding the question about ET intensity, if low intensity and moderate to vigorous intensity are the only possible answers, e.g should be excluded.

Page 3, line 126: “using SAS 9.4” company and city should be added parenthetical.

Page three, line 127-129: The text starting with “also” should be included in the discussion and not in the method section. Please check the spelling in the sentence, there seem to be a word missing.

Table 1 and 2: There are a lot of missing data, total participant inclusion: 5196. The question job position n= 4836, city size n = 4796, education n = 5079, Exercise training = 5012, drinking n = 5193, smoking n = 5193. Based on the missing data, how many subjects are included in the models presented in Table 2- can´t really be 5196

Author Response

In the aim you are writing “Therefore, we investigated the effect of two different shift types (i.e., 3 circuit and 21 circuit) on voluntary participation in ET by Korean firefighters” And in the method “We only included data from firefighters working under 3- and 21 circuit system for the analysis. Also, all of them were outside workers (i.e., firefighting, first aid, rescue) because we outside workers are required to have higher 128 levels of fitness to succeed physically demanding tasks compared to inside workers” Looking at Table 1 you can see the education level and you have explained in text that “High job position includes assistant fire chief, fire chief, above fire sub-deputy chief” Are the high job positions really outside workers?

Thank you for your valuable comment. We have looked closely based on your comment, and we found all of the participants who belonged to high job positions and answered outside workers were working outside of Seoul. It is speculated that there is not enough manpower in such areas and the role boundaries will be unclear compared to Seoul. Furthermore, most of them were working at the safety centers derived from the main fire department in each region. In South Korea, all the firefighters in the safety centers are configured to take on the role of outside. Therefore, participants in high job positions can be included in outside workers in our opinion.

Page 3, line 115: If Yes and No are the only possible answers e.g should be excluded.

Thank you for your suggestion. We changed it based on your opinion.

Page 3, line 117: Regarding the question about ET intensity, if low intensity and moderate to vigorous intensity are the only possible answers, e.g should be excluded.

Thank you for your suggestion. We amended it based on your opinion as well.

Page 3, line 126: “using SAS 9.4” company and city should be added parenthetical.

Thank you for your comment. We changed it based on your comment. The changes are yellow-highlighted.

Page three, line 127-129: The text starting with “also” should be included in the discussion and not in the method section. Please check the spelling in the sentence, there seem to be a word missing.

Thank you for your comment. We changed it based on your comment. The changes are yellow-highlighted.

Table 1 and 2: There are a lot of missing data, total participant inclusion: 5196. The question job position n= 4836, city size n = 4796, education n = 5079, Exercise training = 5012, drinking n = 5193, smoking n = 5193. Based on the missing data, how many subjects are included in the models presented in Table 2- can´t really be 5196

Thank you for your comment. For our analysis, 4,653 data were used. We amended it based on your comment. The changes are yellow-highlighted.
